# Nanobacterial Cellulose from Kombucha Fermentation as a Potential Protective Carrier of *Lactobacillus plantarum* under Simulated Gastrointestinal Tract Conditions

**DOI:** 10.3390/polym15061356

**Published:** 2023-03-08

**Authors:** Sonthirat Charoenrak, Suporn Charumanee, Panee Sirisa-ard, Sittisin Bovonsombut, Ladapa Kumdhitiahutsawakul, Suwalee Kiatkarun, Wasu Pathom-Aree, Thararat Chitov, Sakunnee Bovonsombut

**Affiliations:** 1Department of Biology, Faculty of Science, Chiang Mai University, Chiang Mai 50200, Thailand; sonthirat_charoenrak@cmu.ac.th (S.C.); ladapa.kum@gmail.com (L.K.); wasu.p@cmu.ac.th (W.P.-A.); 2Department of Pharmaceutical Sciences, Faculty of Pharmacy, Chiang Mai University, Chiang Mai 50200, Thailand; suporn.charumanee@cmu.ac.th (S.C.); pmpti008@gmail.com (P.S.-a.); 3Faculty of Engineering and Agro-Industry, Maejo University, Chiang Mai 50290, Thailand; sittinsin@windowslive.com; 4Amazing Tea Limited Partnership (Tea Gallery Group), Chiang Mai 50000, Thailand; suwaleejeed@gmail.com; 5Department of Biology, Research Center of Microbial Diversity and Sustainable Utilization, Faculty of Science, Chiang Mai University, Chiang Mai 50200, Thailand; 6Environmental Science Research Center (ESRC), Faculty of Science, Chiang Mai University, Chiang Mai 50200, Thailand

**Keywords:** nanocellulose, fermented tea, *Lactobacillus plantarum*, bacterial immobilization, biopolymer

## Abstract

Kombucha bacterial cellulose (KBC), a by-product of kombucha fermentation, can be used as a biomaterial for microbial immobilization. In this study, we investigated the properties of KBC produced from green tea kombucha fermentation on days 7, 14, and 30 and its potential as a protective carrier of *Lactobacillus plantarum*, a representative beneficial bacteria. The highest KBC yield (6.5%) was obtained on day 30. Scanning electron microscopy showed the development and changes in the fibrous structure of the KBC over time. They had crystallinity indices of 90–95%, crystallite sizes of 5.36–5.98 nm, and are identified as type I cellulose according to X-ray diffraction analysis. The 30-day KBC had the highest surface area of 19.91 m^2^/g, which was measured using the Brunauer–Emmett–Teller method. This was used to immobilize *L. plantarum* TISTR 541 cells using the adsorption–incubation method, by which 16.20 log CFU/g of immobilized cells was achieved. The amount of immobilized *L. plantarum* decreased to 7.98 log CFU/g after freeze-drying and to 2.94 log CFU/g after being exposed to simulated gastrointestinal tract conditions (HCl pH 2.0 and 0.3% bile salt), whereas the non-immobilized culture was not detected. This indicated its potential as a protective carrier to deliver beneficial bacteria to the gastrointestinal tract.

## 1. Introduction

Cellulose is a carbohydrate polymer found in plant cell walls, algae, fungi, and some bacteria (acetic acid bacteria). Cellulose polymer consists of repeating units of the sugar molecule D-glucopyranose (glucose) connected by β-1,4-glucosidic bonds. The cellulose molecules can be packed together during extended chain conformation via Van der Waals forces and hydrogen bonds to form the basic unit of cellulose fibers, which are, at the nanoscale (1–100 nm), referred to as nanocellulose [1,2,3].

The nanocellulose can be divided into three types, i.e., cellulose nanocrystals (CNCs), cellulose nanofibrils (CNFs), and bacterial nanocellulose (BC), depending on their preparation methods, mechanical features, and physicochemical properties [3]. Bacterial nanocellulose is about 20–100 nm in diameter [4]. It has advantages over plant cellulose in many respects, such as having high purity (free of hemicellulose and lignin), good mechanical properties, high crystallinity and water-holding capacity, and a large surface area. These properties make BC suitable for applications in food, tissue engineering, drug delivery, and the production of polymer nanocomposites. There has been high global demand for nanocellulose for various applications, such as in the paper industry, the biomedical industry, packaging, and wastewater treatment [3]. The global nanocellulose market is expected to expand in size at a compound annual growth rate (CAGR) of 20.0% from 2022 to 2030 [5].

Bacterial nanocellulose is naturally produced by some bacterial species. The first bacterium found to produce cellulose was *Komagataeibacter xylinus* (formerly known as *Gluconacetobacter xylinus*) [6,7]. Since then, the biosynthesis of BC by different groups of microorganisms has been studied. The production of BC through microbial fermentation requires the optimization of various parameters, such as dissolved oxygen, pH levels, and temperature [8].

BC is a great source of novel materials that are used in the food industry. It is a dietary fiber that could serve as a carrier for probiotics. A previous study showed that the probiotic bacterium *Lactobacillus acidophilus* 016 in bacterial cellulose nanofiber developed as a support material had a survival rate of 71% during 24 days of storage at 35 °C [9]. In another study, BC was used to preserve a probiotic *L. plantarum* strain prepared in a ready-to-use form for dairy products, retaining viable cells of 8 log CFU/g (from the initial cell number of 10 log CFU/g) after 5 months of storage at 4 °C [10]. It has also been shown to act as a physical barrier to reduce the deleterious effects of freezing and as a binding matrix for lactic acid bacteria [11,12].

Kombucha bacterial cellulose (KBC) is produced by acetic acid bacteria during the fermentation of kombucha, in which tea and sugar are the main ingredients. KBC is a form of a Symbiotic Culture of Bacteria and Yeast (SCOBY), a type of biofilm that often occurs in multiple layers and increases in thickness with prolonged fermentation time [13,14]. KBC provides a system that protects the microorganisms involved in fermentation from contaminants, thus preserving the bacteria–yeast symbiotic ecosystem that supports their growth and the composition of the kombucha starter culture [15].

The aims of this study were to investigate the properties of KBC from kombucha fermentation and its potential for use as a protective carrier of *Lactobacillus plantarum* TISTR 541 cells in simulated gastrointestinal conditions. The yield of KBC during kombucha fermentation, its physical properties, and the survival of the immobilized strain on KBC after freeze-drying and simulated gastrointestinal tract conditions are reported.

## 2. Materials and Methods

### 2.1. Kombucha Bacterial Cellulose Production and Purification

Green tea and kombucha starter were obtained from Amazing Tea Limited Partnership, Tea Gallery Group (Thailand) Co., Ltd., Chiang Mai, Thailand. The green tea kombucha used in this study was prepared using the method routinely used by the company. Green tea leaf was infused for 30 min in 150 L of boiling water at a concentration of 1% (*w/v*). The tea infusion was filtered into a 200 L container, and 10% (*w/v*) sucrose was added. After that, 10% (*v/v*) of the kombucha starter culture from a previous batch was inoculated. The kombucha tea was prepared in three treatments (each in triplicate), which were incubated for 7, 14, and 30 days at room temperature (30 ± 2.0 °C).

The KBC that formed on the surface of the kombucha broth was collected at the end of the fermentation time for each treatment, i.e., on days 7, 14, and 30. The fresh KBC was measured for wet weight (WW) using an analytical balance (PL602-L, Mettler Toledo, Zurich, Switzerland). It was then dried in a hot air oven at 60 °C for 3–5 days until a constant weight was achieved, and dry weight (DW) was subsequently recorded. The water content of cellulose pellicles was determined according to the method of Orlando et al. [16]. The KBC yields (%) were calculated as follows [17]:KBC Yield (%) = (KBC dry weight (g)/amount of sucrose (g)) × 100(1)

KBC purification was modified from the method described by Sheykhnazari et al. [18]. In brief, the KBC sheets harvested on days 7, 14, and 30 were cut into pieces and washed with distilled water. The KBC pieces were then boiled in distilled water at 90 °C for 2 h, followed by boiling in a 0.5 M NaOH solution for 15 min at 90 °C to remove impurities. The KBC was then washed several times with distilled water until a neutral pH was achieved. The purified KBC was dried in an oven at 60 °C until the weight was constant (3–5 days). It was then ground into powder using a blender (La Moulinette DPA 130, Tefal, France) and sieved through a No. 60 sieve in a vibratory sieve shaker (AS 300 control, Retsch, Haan, Germany) to retrieve powder with an average particle size of 250 microns. The KBC powder was sterilized for 15 min at 121 °C in an autoclave and dried at 60 °C for 3 days.

The kombucha broth was sampled on days 7, 14, and 30 of fermentation in triplicate to analyze the pH, acetic acid content, and microbial count (yeasts and acetic acid bacteria). The pH level was measured using an electronic pH meter (F20, Mettler-Toledo, Schwerzenbach, Switzerland). Total acetic acid content was determined by titrating the kombucha broth with a 0.5 M NaOH solution, and the acetic acid concentration was then calculated in grams per liter. The cell counts of yeasts and acetic acid bacteria in the kombucha were determined using the drop plate technique [19]. For the yeast count, 10 mL of kombucha broth was serially diluted with 0.1% (*w/v*) peptone water and dropped onto dichloran rose bengal chloramphenicol (DRBC) agar (Merck, Darmstadt, Germany). The DRBC plates were then incubated at 25 °C for 5 days. The number of acetic acid bacteria was determined on Hestrin–Schramm (HS) agar supplemented with 50 ppm cycloheximide to inhibit yeasts and molds. The HS agar plates were incubated at 30 °C for 3 days. After incubation, colonies of yeasts and acetic acid bacteria were counted and calculated in colony-forming units per milliliter of kombucha.

### 2.2. Analysis of Kombucha Bacterial Cellulose Properties

#### 2.2.1. Morphological Structure

The morphology and the surface structure of KBC were analyzed using a scanning electron microscope (SEM). To prepare a sample for SEM analysis, the KBC powder was mounted on a double-sided adhesive carbon tape and excess KBC powder was then removed. The tape mounted with the sample was placed on a stub and dried in a hot air oven at 60 °C for 24 h. After that, the samples were coated with a thin layer of gold. The KBC fibrils were examined under SEM (JSM-5910LV, JEOL Technic Ltd., Tokyo, Japan) with an accelerating voltage of 15 kV [20].

#### 2.2.2. Cellulose Type and Crystallite Size

The type and crystallite size of the KBC were analyzed using the X-ray diffraction (XRD) technique (performed using a Miniflex II desktop X-ray diffractometer (Rigaku, Tokyo, Japan) from the Science and Technology Service Center, Faculty of Science, Chiang Mai University). The X-ray diffraction patterns were recorded and used to calculate the crystallinity index (CrI) and crystallite size (CrS). The CrI was determined by comparing the height of the (200) peaks (I200, 2q = 22.6°) to the height of the minimum point (Iam, 2q = 18°) between the (200) peak and the (I110, 2q = 14°) peak [21].
CrI (%) = (1 − (I_am_/I_200_)) × 100(2)

#### 2.2.3. Determination of Surface Area, Pore Volume, and Pore Size

The surface area and porosity of the KBC samples were analyzed using the Brunauer–Emmett–Teller (BET) equation. The KBC powder samples (0.06 g), harvested on days 7, 14, and 30, were subjected to a Brunauer–Emmett–Teller surface analyzer (Model autosorb1 MP, Quantachrome Instruments, Boynton Beach, FL, USA) with nitrogen gas as an absorbate. All samples were vacuum-degassed at room temperature for 24 h prior to analysis [9].

### 2.3. Immobilization of Lactobacillus plantarum TISTR 541 Cells on Kombucha Bacterial Cellulose

#### 2.3.1. *Lactobacillus plantarum* Strain

*L. plantarum* TISTR 541, a representative of beneficial bacterial strains, was used for immobilization on KBC. This strain was provided by the Microbiological Resource Centre, Thailand Institute of Scientific and Technological Research (TISTR), Thailand. The strain was preserved as a frozen glycerol stock culture in de Man–Rogosa–Sharpe (MRS) broth (Difco, Detroit, MI, USA) with 20% (*v/v*) glycerol.

#### 2.3.2. Preparation of *L. plantarum* for Immobilization

The glycerol culture was activated in MRS broth and incubated at 37 °C for 24 h under anaerobic conditions (using an anaerobic jar with an Anaerocult A pad (Merck, Darmstadt, Germany)). This culture was then transferred to 250 mL of MRS broth contained in a tightly sealed 500 mL flask and incubated at 37 °C for 24 h. The cells of *L. plantarum* TISTR 541 were then collected by centrifugation at 7000 rpm at 4 °C for 15 min. The cell pellets were washed three times with 0.85% (*w/v*) NaCl solution and centrifuged at 7000 rpm at 4 °C for 15 min. The cells were resuspended in 10 mL of 1.0% (*w/v*) inulin solution, and this cell suspension was used for immobilization on KBC powder (day 30). The number of *L. plantarum* was determined using the drop plate method [19].

#### 2.3.3. Immobilization

The cell suspension of *L. plantarum* TISTR 541 was prepared in 10 mL of 1.0% (*w/v*) inulin solution. This cell suspension had a *L. plantarum* concentration of approximately 16 log CFU/mL. The suspension was transferred to a tube containing 1.0 g of sterilized KBC powder and mixed using a vortex mixer. Immobilization of cells was carried out using the adsorption–incubation method [12], in which the mixture was placed in a shaker operated at 120 rpm for 6 h and then incubated at 37 °C for 18 h. 

#### 2.3.4. Enumeration of Immobilized Cells on KBC

The number of immobilized cells on KBC was determined after incubation. Samples were washed with phosphate buffered saline (PBS, Sigma-Aldrich, St. Louis, MO, USA) to remove free cells. The cellulose was then digested with cellulase (100 μL/mL in 0.05 M citrate buffer (pH 4.8), Sigma-Aldrich, St. Louis, MO, USA) [22]. The number of immobilized cells was determined by serial dilution with 0.1% (*w/v*) peptone water was then dropped onto MRS agar. The MRS agar plates were incubated for 48 h at 37 °C under anaerobic conditions (using an anaerobic jar with an Anaerocult A pad (Merck, Darmstadt, Germany)). The total number of bacteria in log CFU/g of cellulose was calculated [12].

#### 2.3.5. Visualization of Immobilized *L. plantarum* Cells on KBC under SEM

The KBC pellicles with immobilized *L. plantarum* TISTR 541 cells were mounted and coated with gold, as in Section 2.2.1, before being examined using a JEOL JSM-5910LV microscope (JEOL Technic Ltd., Tokyo, Japan) with an accelerating voltage of 15 kV [20].

### 2.4. Investigation of the Tolerance of Immobilized L. plantarum Cells to Acid and Bile Salts

The KBC immobilized with *L. plantarum* was dried using the freeze-drying technique. The immobilized *L. plantarum* after freeze-drying were enumerated using the drop plate technique. The survival of the immobilized *L. plantarum* cells under simulated gastrointestinal (GI) tract conditions was determined as described by Fijalkowski et al. [12].

To investigate the effect of simulated GI tract conditions on the survival of the immobilized *L. plantarum* cells, 100 mg of KBC immobilized with *L. plantarum* cells was added to 10 mL of MRS broth that was adjusted to pH 2.0 with HCl, mixed, and incubated at 37 °C for 2 h. The treated immobilized cells were then collected after centrifugation at 7000 rpm for 15 min, and the culture supernatant was discarded. After that, 10 mL of MRS broth containing 0.3% (*w/v*) bile salts was added to the cells, mixed, and further incubated at 37 °C for 4 h. The number of surviving *L. plantarum* cells after being in an acidic condition (pH 2.0) and after being exposed to 0.3% bile salts was determined using the drop plate technique [19] after incubation under anaerobic conditions at 37 °C for 48 h.

## 3. Results

### 3.1. Bacterial Cellulose Production from Kombucha

Kombucha bacterial cellulose (KBC) from kombucha prepared from green tea was harvested on days 7, 14, and 30 from the separate fermentation tank (set in triplicate) (2.1). The cellulose layers formed on the surface of kombucha gradually increased as fermentation progressed (see an example in Figure 1a). The fresh KBC obtained in this study was light yellow to brown and had a homogeneous texture (Figure 1b). 

The KBC portion was collected for the measurement of wet weight. It was then dried (as in Section 2.1) (Figure 1c), and the dry weight was measured. The dried KBC powder was prepared (Figure 1d), and its physical properties were determined. The kombucha broth portion from each treatment was also collected for analysis of pH, total acid concentration, and the amount of acetic acid bacteria and yeasts. The results from these analyses are shown in Table 1.

The wet weight of KBC increased from 1,162 g to 3,931 g and the dry weight from 120 g to 880 g from day 7 to day 30 of fermentation. The KBC yield also increased from 0.9% to 6.5%. On day 7, when it was a thin film, the KBC had a high moisture content of 89.7%. When it developed into thicker layers with longer fermentation time, on days 14 and 30, the KBC had lower moisture content of 76.5% and 77.6%, respectively.

The tea infusion had an acidic pH (4.6 ± 0.1), and the pH dropped to 3.6 after the addition of kombucha starter. As the fermentation progressed, the pH dropped until it reached 2.3 on day 30. The acetic acid concentration increased throughout the course of fermentation and was highest on day 30, with a concentration of 19.50 g/L observed (from 1.44 g/L on day 1). Yeasts and acetic acid bacteria in kombucha broth were enumerated on DRBC and HS agar, respectively. The results are shown in Table 1.

There were also microbiological changes in kombucha during the fermentation process. At the beginning of fermentation, after the addition of the kombucha starter, the yeast count was 5.0 log CFU/mL. The yeast counts were highest on days 7 and 14 and decreased on day 30 of fermentation. In addition, the number of acetic acid bacteria in the kombucha broth increased from 3.6 log CFU/mL at the beginning of fermentation to 7.8 and 7.5 log CFU/mL on days 7 and 14, increasing by approximately 4 log cycles. The count dropped approximately 3 log cycles to 4.2 log CFU/mL on day 30. 

### 3.2. Morphological Structure and Physical Properties of Kombucha Bacterial Cellulose

#### 3.2.1. Morphological Structure of Kombucha Bacterial Cellulose

KBC pellicles harvested from kombucha on days 7, 14, and 30 were analyzed for KBC properties. The morphological surface structures of the KBC powder prepared from KBC collected at different fermentation periods and analyzed using SEM are shown in Figure 2. A thin network of cellulose fibrils covering the cells of acetic acid bacteria was already observable in KBC on day 7 (Figure 2a). The random nanofibrils around the cellulose-producing bacteria accumulated and appeared denser on day 14, with additional layers forming around the rod-shaped bacterial cells entrapped in the network (Figure 2b). By day 30, the KBC layers were clearly visible; the surface was rougher, while the outer layer of cellulose fibrils remained intact in the network structure (Figure 2c). 

#### 3.2.2. Cellulose Type and Crystallite Size of KBC 

X-ray diffraction (XRD) measurements were performed to determine the crystalline structure of KBC obtained from kombucha on days 7, 14, and 30 of fermentation. The XRD peaks in Figure 3 indicated that the KBC had intensities of 110 and 200 diffractions. Two distinct peaks at two angles were observed for all treatments. The peaks at 2θ of 14.58°, 14.18°, and 14.52°, corresponding to the primary diffraction (Iα(110)), were detected on days 7, 14, and 30 of fermentation, respectively. Furthermore, peaks at 2θ of 22.58°, 22.38°, and 22.46°, which corresponded to the secondary diffraction (Iβ(200)), were observed for KBC on days 7, 14, and 30, respectively. The presence of these peaks indicates that the KBC contains the typical crystalline forms of type I cellulose.

The crystallinity index for each KBC sample was calculated based on the peak intensity data obtained by the Segal method [21]. The crystallinity indices of 90%, 95%, and 91% were observed for KBC on days 7, 14, and 30, respectively. Although the crystallinity index of KBC harvested on day 14 was slightly higher than that of days 7 and 30, there were no significant differences according to Tukey’s test (*p*-value < 0.05).

Crystallite size was calculated from the XRD diffractogram based on peak intensity data using Scherrer’s method [21]. The crystallite sizes of KBC at days 7, 14, and 30 of fermentation were 5.36, 5.94, and 5.98 nm, respectively, which were not significantly different from each other (*p*-value < 0.05; Tukey’s test). This suggests that the crystallite sizes of KBC remained relatively constant during the observation period and did not change with the age of KBC.

#### 3.2.3. Surface Area, Pore Volume, and Pore Size of Kombucha Bacterial Cellulose

The surface area and pore volume of KBC are very important factors, especially when it is being considered as a potential material for bacterial immobilization. The surface areas, pore volumes, and pore sizes of the KBCs analyzed using the Brunauer–Emmett–Teller (BET) equation are listed in Table 2. The average pore sizes of the KBCs were larger with longer fermentation time; however, the pore volumes and surface areas were not proportionally related to fermentation time.

### 3.3. Immobilization of L. plantarum TISTR 541 Cells on Kombucha Bacterial Cellulose

Purified powdered KBC produced from kombucha on day 30 was used for the immobilization of *L. plantarum* TISTR 541 cells through the adsorption–incubation method (2.3.3). For immobilization, a suspension of fresh *L. plantarum* TISTR 541 cells in inulin was prepared at a concentration of 16.18 ± 0.01 log CFU/mL. The immobilized *L. plantarum* cells on KBC were incubated for another 24 h. After incubation, immobilized *L. plantarum* cells were found at 16.20 ± 0.03 log CFU/g KBC. 

The immobilized *L. plantarum* cells on KBC were further prepared into a freeze-dried form, which can be easily applied to health products. After freeze-drying, the immobilized *L. plantarum* cell concentration was reduced to 7.98 ± 0.19 log CFU/g, whereas the non-immobilized culture was reduced much further to 5.20 ± 0.02 log CFU/g (Table 3). 

The immobilized and freeze-dried *L. plantarum* TISTR 541 cells on KBC were visualized using SEM. The SEM images showed that the cellulose fibrils in KBC were tightly packed (Figure 4a), with *L. plantarum* cells (rod-shaped cells) attached to the KBC surface that were randomly spread around the KBC surface structure (Figure 4b,c).

### 3.4. Tolerance of Immobilized Lactobacillus plantarum TISTR 541 Cells to Acid and Bile Salts

The ability of KBC to act as a protective carrier of *L. plantarum* TISTR 541 cells was investigated. After the freeze-drying process, the number of *L. plantarum* TISTR 541 cells immobilized on KBC decreased compared to the non-freeze-dried immobilized culture (3.3). The survival of immobilized freeze-dried *L. plantarum* TISTR 541 cells under acidic conditions (pH 2.0) and in a solution containing 0.3% bile salt was then investigated. The number of immobilized *L. plantarum* cells decreased by approximately 2.82 log CFU/g after being exposed to the acidic conditions for 2 h, and further decreased by 2.22 log CFU/g after being exposed to 0.3% bile salts for 4 h (Table 3). This resulted in a final level of viable *L. plantarum* cells on KBC of 2.94 ± 0.21 log CFU/g, which was reduced by 5.04 log CFU/g from the initial number (freeze-dried immobilized culture). This number represents an expected number of live bacteria in the gastrointestinal tract after being administered in a freeze-dried and immobilized form.

## 4. Discussion

In this study, bacterial cellulose harvested from green tea kombucha was investigated for its properties and its potential to be used as a protective carrier of L. plantarum TISTR 541, a representative strain of beneficial bacteria. Kombucha bacterial cellulose (KBC) is also known as a Symbiotic Culture of Bacteria and Yeasts (SCOBY). KBC is considered a by-product in kombucha fermentation and is a nanopolymer produced by cellulose-producing acetic acid bacteria. The properties of kombucha broth and KBC were investigated on days 7, 14, and 30 of fermentation.

For kombucha, the physical and microbiological properties were determined (such as pH, total acid concentration, and the number of yeasts and acetic acid bacteria), since these properties depend on the raw materials and fermentation conditions used in kombucha production. There is a general tendency for the pH of kombucha broth to shift from a neutral pH to an acidic pH during kombucha fermentation [23,24]. In our experiment, the pH of the initial green tea infusion was slightly acidic (pH 4.6), and it immediately dropped to 3.6 after the addition of the kombucha starter. This initial acidic pH value is important to prevent the growth of spoilage and pathogenic microbial contaminants [25]. The kombucha pH values gradually decreased to 2.3 by day 30 of fermentation and were inversely correlated with acetic acid concentration, which increased from 1.4 to 19.5 g/L. Our results were similar to those from other studies, in which the pH values of kombucha decreased (from 5.03 to 1.88 after 21 days of black tea kombucha fermentation) and acetic acid concentration increased (from 0.65 to 16.57 g/L) [24]. The acidic pH values that dropped to nearly 2.0 in our study still seemed to support the growth of acetic acid bacteria and their cellulose production in green tea kombucha.

The number of yeasts and acetic acid bacteria was also investigated in the kombucha broth. The number of yeasts increased from approximately 5 log CFU/mL to the highest levels (*ca.* 7 log CFU/mL) on days 7 and 14, after which it dropped to 6.4 log CFU/mL on day 30 of fermentation. The same phenomenon occurred with acetic acid bacteria, which also increased to the highest levels of approximately 8 log CFU/mL on days 7 and 14 before then dropping to 4.3 log CFU/mL on day 30. This correlated with the drastic decrease in pH and increase in acetic acid concentration observed on day 30. The increase in the number of yeasts and acetic acid bacteria during the first two weeks of fermentation was due to the metabolism of sucrose in the green tea infusion to glucose and fructose by yeasts in the kombucha starter, thus producing ethanol. The ethanol was oxidized by acetic acid bacteria to organic acids [23], largely acetic acid, which corresponded to the gradual increase in acid concentration until day 14. The exhaustion of the carbon source, the low pH, and the high acidity in the kombucha eventually affected the growth and survival of yeasts and acetic acid bacteria, resulting in a decrease in their numbers when fermentation was prolonged to 30 days. The decrease in acetic acid bacteria was due to acid shock, likely caused by a low pH and the high organic acid concentration [23]. This finding was similar to that previously reported by Chen and Liu [26], who found that the acetic acid bacterial count in black tea kombucha rose from approximately 4 log CFU/mL to 6.2 log CFU/mL on day 6 before then decreasing to 5.2 log CFU/mL after 30 days. However, the remaining acetic acid bacteria continued to produce acetic acid after day 14, resulting in the accumulation of acetic acid in the kombucha broth and leading to the highest acetic acid concentration on day 30. Ayed et al. reported that yeasts and acetic acid bacteria could grow by assimilating the sugar present in the juice, as demonstrated by their increasing cell numbers in broth for up to 6 days of culture and their decreasing number thereafter through to the end of fermentation [27]. The increase in acetic acid concentration during the later stage of fermentation, even after the acetic acid bacteria counts dropped, was also observed by Jayabalan et al. for their study in green tea kombucha [23]. This indicated that although their numbers were not at their highest levels at the later stage of kombucha fermentation, both yeasts and acetic acid bacteria could still survive and be active in such acidic conditions; they could also utilize metabolites formed during the fermentation process, as previously noted by other researchers [26,28,29].

As for the KBC produced from the green tea kombucha in this study, there was a gradual increase in KBC yield that was related to fermentation time. The maximum KBC yield of 6.5% was achieved on day 30 of fermentation. The KBCs harvested at different fermentation times were characterized using a combination of techniques, such as SEM, XRD, and BET analyses. SEM showed that the cellulose layer is composed of a compact network of overlapping layers of fibrils covering bacterial cells. The structure of cellulose harvested at different times differed slightly. By day 7, the KBC was formed as a thin layer covering the aggregated cells of acetic acid bacteria, which is the typical structure of young kombucha cellulose fibrils [21]. As the fermentation time passed, the layers thickened, and the fibrils became denser. By day 30, multiple layers were randomly formed, and this could explain the larger surface area of the KBC from day 30 than that from day 14. KBC fibers have been recognized as nanofibers. Tapias et al. [30] found that the KBC nanofibers obtained from six different combinations of herbal tea infusions had diameters in the range of 20–100 nm. The structure and the nanoscale sizes of KBC contribute to its high surface area and high porosity.

The XRD analysis revealed that the bacterial cellulose produced from the green tea kombucha on days 7, 14, and 30 exhibited peaks at 2θ angles of approximately 14° and 22°, which corresponded to the (−110) and (200) crystal planes, respectively, demonstrating the typical crystalline structure of type I cellulose [31]. The crystallinity index (90–95%) for KBC on days 7, 14, and 30 indicated cellulose with a high degree of purity. According to a study by Leonarski et al., the production of a kombucha-like beverage and bacterial cellulose using an acerola by-product as a raw material resulted in bacterial cellulose with high crystallinity that ranged from 81.4% to 96.7% [32]. The degrees of crystallinity of different types of bacterial cellulose reported in other studies were in the range of 71–92% [25,33,34,35,36,37,38,39,40].

Purity and crystallinity are crucial features for enhancing the mechanical properties of bacterial cellulose [41]. The crystallite size of KBC in our study was in the range of 5.3–5.9 nm, similar to the average crystallite size of KBC from black tea kombucha reported in a previous study, which was 5.2 nm [40]. However, in our study, no significant difference in crystalline structure and crystallinity index was observed among KBCs obtained after different durations of fermentation. Such nanoscale crystal sizes make bacterial cellulose a suitable biopolymer for application as a delivery agent [42]. Thomas et al. formulated a new hybrid polymer by combining nanosized alginate and cellulose nanocrystals. This formulation is suitable for the controlled oral delivery of rifampicin and led to a better outcome in treating *Mycobacterium tuberculosis* [43]. In food preservation, packaging films for frankfurters were made of bacterial cellulose containing nisin as an antimicrobial agent. These films were found to prevent the growth of *Listeria monocytogenes* and total aerobic bacteria [44].

The pore volume, pore size, and specific surface area of KBCs were determined using BET analysis. Pore size and pore size distribution can be used to evaluate a material’s mechanical properties and its ability to adsorb other molecules. In this study, average KBC pore sizes seemed to be positively correlated to fermentation time; the longer the fermentation time, the larger the pore size. The average pore sizes of the KBCs found in our study (4.8–6.1 nm) are within the range of bacterial cellulose pores (2–50 nm), which are classified as mesopores [9]. The increase in KBC pore size during kombucha fermentation could occur because of the dynamic growth of kombucha microorganisms and the continuous accumulation of cellulose networks. As the additional layers of fibers are added to the membrane, they become less tightly packed and less orderly arranged than an accumulated single layer, or even multilayers that are still held together in one unit. This could have resulted in more crevices forming in the cellulose sheet as fermentation continued and the KBC aged, which in turn would have increased the average pore size. Additionally, the polysaccharides produced by bacteria can create a surrounding matrix, which can increase how uneven the surfaces are and could possibly separate the cellulose fibers by creating larger spaces between them. The results from a previous study by Ramire-Carmona et al. are well supported by electron micrographs and show that these polysaccharides could be the key factor contributing to the increase in KBC pore size with increased fermentation time [45].

Surface area, unlike pore size, was not proportionally correlated to fermentation time. The surface areas of the KBCs on days 7 and 30 were found to be greater than those on day 14. This could be because of the thin sheet structure of KBC formed during the early stage of fermentation and the multiple layers formed during the later fermentation periods. The surface area of KBC on day 30 was 19.91 m^2^/g, which was considered high compared with that of bacterial cellulose produced from a pure culture of *Acetobacter* in modified HS medium (4.2 m^2^/g), which was used for bacterial immobilization [9].

Bacterial cellulose has been recognized by the US FDA as a safe food additive since 1992 [46]. It has various food applications, such as use in traditional desserts, low-cholesterol diets, food and beverage additives, and packaging [47]. To evaluate its potential use as a carrier of probiotics or beneficial bacterial cells, the purified cellulose produced from kombucha on day 30 was used to immobilize *L. plantarum* TISTR 541 cells using the adsorption–incubation method. The results showed that the KBC had a high capability to contain bacterial cells. It could accommodate up to 16.57 log CFU/g of *L. plantarum* cells after immobilization. After freeze-drying, which is the chosen process to transform the immobilized bacterial strain into a ready-to-use form, the number of *L. plantarum* TISTR 541 cells on KBC decreased to about 8 log CFU/g. This is, however, still at an acceptable level for its further intended application. Although the immobilized *L. plantarum* strain further decreased in the simulated GI tract conditions tested (acidic pH (pH 2.0) followed by 0.3% bile salt), the cells still partly survived, while without KBC as a protective barrier, the viable cells of *L. plantarum* TISTR 541 could not be recovered. The results of this study show that the immobilization method used in this study allowed for the preparation of a high concentration of immobilized *L. plantarum* cells on KBC, which could still be retained in a sufficient amount after freeze-drying. It was also sufficient to partially protect the bacterial cells against the harmful effect of HCl, suggesting that KBC could be used as an immobilizing material to deliver beneficial bacteria to the gastrointestinal tract. The potential use of KBC as a supporting material for probiotic strains was also previously proposed. Jayani et al. found that the adsorption–incubation technique was highly advantageous as it retained the highest levels of viable *L. acidophilus* 016 cells on BC fibers [9]. Another study by Savedboworn et al. showed the high survival of freeze-dried *L. plantarum* cells in skim milk after exposure to simulated gastric juice [48]. Fijalkowski et al. also reported that the fibrous structure of bacterial cellulose in its wet form could also prevent the destructive effects of freezing and provide an attachment matrix for lactic acid bacteria [12]. Cryoprotectant agents of appropriate types and concentrations can improve freeze-drying survival [11,12].

In terms of its effect on health, bacterial cellulose is a dietary fiber that benefits digestive health and can reduce the risk of chronic diseases such as obesity, diabetes, and cardiovascular disease [49]. It can also be used as an additive in low-calorie foods [50]. Considering these functions, together with the ability of KBC to act as a protective carrier of *L. plantarum*, as demonstrated in this study, this by-product from kombucha fermentation is recommended as a functional bio-protective carrier for beneficial or probiotic bacteria, which has a high potential in various food and pharmaceutical applications.

## 5. Conclusions

Kombucha bacterial cellulose (KBC) produced from green tea kombucha was harvested on days 7, 14, and 30 of fermentation. Their physical properties were investigated, as well as their potential for use as an immobilizing material and a protective carrier of *L. plantarum* cells. The KBC yields were proportional to fermentation time; the highest yield was obtained on day 30. Through SEM, the KBC was seen as a network of random fibrils covering the cellulose-producing bacteria present in kombucha. KBC developed from a thin monolayer to multiple layers as fermentation time increased. The crystallinity indices of KBC were 90–95%, indicating its high purity. The pore sizes of the KBCs were in the range of 4.865–6.189 nm, and surface area ranged from 12.16–19.91 m^2^/g. The KBC produced on day 30, which had the highest surface area, was used to immobilize *L. plantarum* TISTR 541 cells via the adsorption–incubation method, which could retain up to 16.57 log CFU/g of *L. plantarum* cells. Although the number decreased after freeze-drying, the amount of *L. plantarum* cells on KBC still remained at a sufficient level for further application in health products (*ca*. 8 log CFU/g). KBC, to a certain degree, could protect *L. plantarum* cells from simulated GI tract conditions (an acidic pH of 2.0 and the presence of 0.3% bile salt). It is therefore suggested as a potential protective carrier of beneficial bacterial cells that are to be delivered to the gastrointestinal tract.

## Figures and Tables

**Figure 1 polymers-15-01356-f001:**
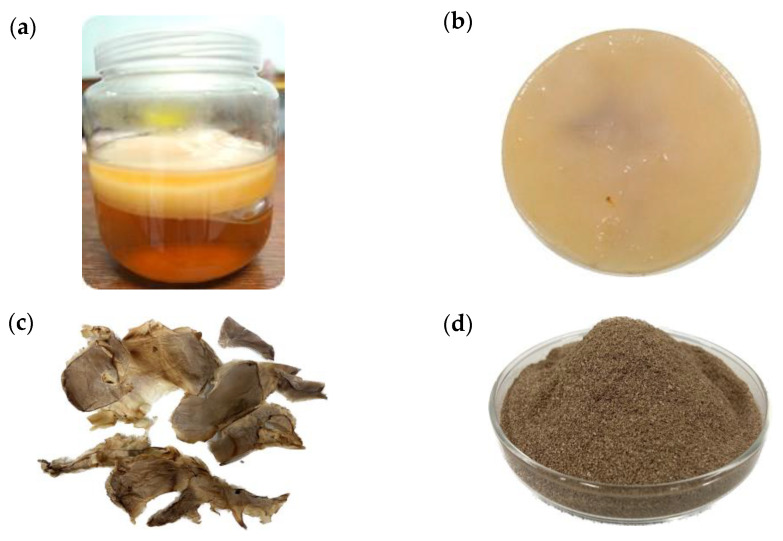
Physical characteristics of kombucha bacterial cellulose (KBC): (**a**) the KBC layer formed after fermentation of kombucha for 30 days; (**b**) the top-view appearance of KBC; (**c**) dry KBC pieces that were thinly sliced from KBC harvested on day 30 of fermentation; and (**d**) KBC powder prepared from KBC harvested on day 30 of fermentation. The ground KBC was sieved through a No. 60 sieve, providing a fine cellulose powder of approximately 250 µm.

**Figure 2 polymers-15-01356-f002:**
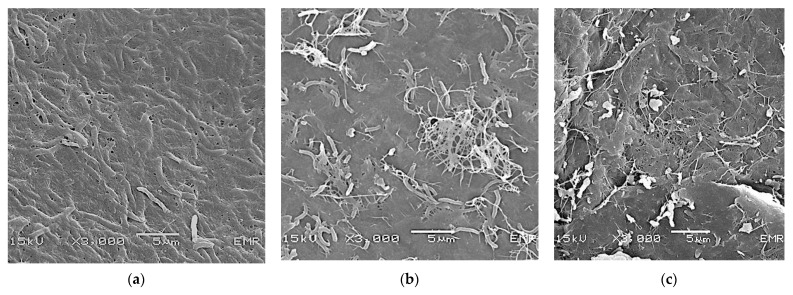
SEM micrographs of kombucha bacterial cellulose produced by acetic acid bacteria under different fermentation times: (**a**) day 7; (**b**) day 14; and (**c**) day 30.

**Figure 3 polymers-15-01356-f003:**
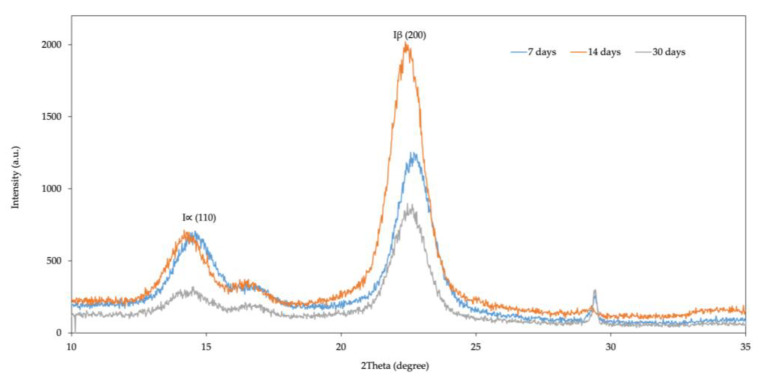
X-ray diffraction analysis of bacterial cellulose produced from kombucha during fermentation on days 7, 14, and 30.

**Figure 4 polymers-15-01356-f004:**
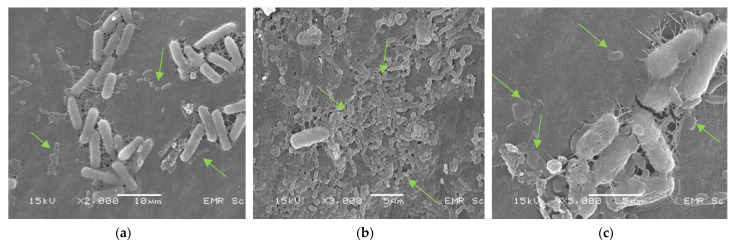
Electron micrographs of *Lactobacillus plantarum* TISTR 541 cells immobilized on KBC (day 30) using the adsorption–incubation method after freeze-drying. (**a**); *L. plantarum* attached on KBC fibrils at 2000×, (**b**); *L. plantarum* cells attached on KBC fibrils at 3000×, and (**c**); *L. plantarum* cells attached on KBC fibrils at 5000×.

**Table 1 polymers-15-01356-t001:** Yields and basic properties of kombucha bacterial cellulose and kombucha broth at different times of fermentation.

Time of Fermentation (Days)	Kombucha Bacterial Cellulose				Kombucha Broth			
	WW (g)	Moisture Content (%)	DW (g)	Yield of DW (%)	pH	Acetic Acid Conc. (g/L)	Number of Microorganisms (log CFU/mL)	
							Yeasts	AAB *
0	-	-	-	-	3.6 ± 0.1	1.44 ± 0.08	5.1 ± 0.4	3.6 ± 0.1
7	1162 ± 28	89.7	120 ± 10	0.9	3.2 ± 0.0	3.42 ± 0.08	7.0 ± 0.5	7.9 ± 0.1
14	2258 ± 19	76.5	530 ± 21	3.9	3.0 ± 0.0	6.24 ± 0.33	7.1 ± 0.1	7.5 ± 0.4
30	3931 ± 43	77.6	880 ± 27	6.5	2.3 ± 0.0	19.50 ± 0.58	6.4 ± 0.2	4.3 ± 0.2

The figures are presented as means ± SD. * AAB: acetic acid bacteria.

**Table 2 polymers-15-01356-t002:** Pore characteristics and surface area of bacterial cellulose produced from kombucha.

Time (Day)	Total Pore Volume (cc/g)	Average Pore Size (nm)	Surface Area (m^2^/g)
7	0.0451	4.865	18.52
14	0.0345	5.674	12.16
30	0.0616	6.189	19.91

**Table 3 polymers-15-01356-t003:** Survival of *Lactobacillus plantarum* TISTR 541 cells under acidic conditions and in a bile salt solution.

Form of *L. plantarum* Culture	Initial No. of *L. plantarum* Cells before Freeze-Drying (log CFU/mL)	No. of *L. plantarum* Cells after Freeze-Drying (log CFU/g)	Survival of *L. plantarum* Cells after Being Exposed to Simulated GI Tract Conditions			
			MRS (pH 2.0)		MRS with 0.3% (*w/v*) Bile Salt	
			Number (log CFU/g)	Log Reduction *	Number (log CFU/g)	Log Reduction *
Non-immobilized culture	15.55 ± 0.12	5.20 ± 0.02	ND **	-	ND **	-
Immobilized culture on KBC	16.20 ± 0.03	7.98 ± 0.19	5.16 ± 0.62	2.82	2.94 ± 0.21	5.04

* calculated from mean value of the number of viable *L. plantarum* cells in the freeze-dried culture; ** not detected by the drop plate method used in this study.

## Data Availability

Not applicable.

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
