# Peer review of "Nanobacterial Cellulose from Kombucha Fermentation as a Potential Protective Carrier of Lactobacillus plantarum under Simulated Gastrointestinal Tract Conditions"

_polymers, 2023, doi:10.3390/polym15061356_

Round 1
Reviewer 1 Report
Dear Authors,
My comments below:
Lines 235-241: Conclusions, comparisons to other works are missing.
Lines 270-274: Could you give a comment please? What these data mean?
Lines 280-281 and in the methodology: This is an equation and I would rather say using an equation instead of using a method.
Line 333: Concentration is a physical property, not chemical one.
Line 351: The same instead of this same
Line 404: Could you give some examples of those delivery agents please?
Author Response
"Please see the attachment."

Reviewer 2 Report
The article «Nanobacterial Cellulose from Kombucha Fermentation as Potential Protective Carrier of Lactobacillus plantarum under Simulated Gastrointestinal Tract Conditions» is attracted to a relevant topic and has a high applied value.
Authors showed, that kombucha bacterial cellulose can be used as a biomaterial for microbial immobilization. In this study, authors investigated the properties of kombucha bacterial cellulose and it potential as a carrier for Lactobacillus plantarum. So, this article demonstrated potential of kombucha bacterial cellulose as a protective carrier to deliver beneficial bacteria to the gastrointestinal tract.
The text of article is well structured, written in sufficient detail and logically. The authors conducted an extensive experiment.
However, a number of minor questions arise:
1. Why the intensity of XRD pattern of 30-day-fermentated BC is lower compared other samples? May be this correlate with pore size?
2. Why KBC pore sizes rise with an increase in fermentation time?
Author Response
"Please see the attachment."
